# Fair Graph Representation Learning via Sensitive Attribute Disentanglement

## ABSTRACT

Group fairness for Graph Neural Networks (GNNs), which emphasizes algorithmic decisions neither favoring nor harming certain groups defined by sensitive attributes (e.g., race and gender), has gained considerable attention. In particular, the objective of group fairness is to ensure that the decisions made by GNNs are independent of the sensitive attribute. To achieve this objective, most existing approaches involve eliminating sensitive attribute information in node representations or algorithmic decisions. However, such ways may also eliminate task-related information due to its inherent correlation with the sensitive attribute, leading to a sacrifice in utility. In this work, we focus on improving the fairness of GNNs while preserving task-related information and propose a fair GNN framework named **FairSAD**. Instead of eliminating sensitive attribute information, FairSAD enhances the fairness of GNNs via Sensitive Attribute Disentanglement (SAD), which separates the sensitive attribute-related information into an independent component to mitigate its impact. Additionally, FairSAD utilizes a channel masking mechanism to adaptively identify the sensitive attribute-related component and subsequently decorrelates it. By leveraging SAD, FairSAD not only improves fairness but also uncovers the latent factors underlying real-world graph-structured data, thereby preserving task-related information. Furthermore, experiments conducted on several real-world datasets demonstrate that FairSAD outperforms other state-of-the-art methods by a significant margin in terms of both fairness and utility performance. Our source code is available at https://anonymous.4open.science/r/FairSAD/.

## CCS CONCEPTS

• **Computing methodologies** → **Machine learning**.

## KEYWORDS

Graph Neural Networks, Group Fairness, Graph Representation Learning

**ACM Reference Format:**
Anonymous Author(s). 2018. Fair Graph Representation Learning via Sensitive Attribute Disentanglement. In *Proceedings of Make sure to enter the correct conference title from your rights confirmation emai (Conference acronym 'XX).* ACM, New York, NY, USA, 11 pages. https://doi.org/XXXXX.XXXXX

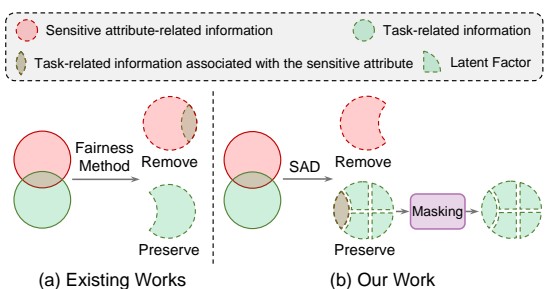

**Figure 1: Comparison of FairSAD with existing works. Existing works inevitably eliminate the task-related information due to its correlations with the sensitive attribute.**

## 1 INTRODUCTION

Graph Neural Networks (GNNs) have emerged as a powerful tool for learning node representation from graph-structured data, which are employed in various applications such as recommendation systems [13], and protein-protein interaction prediction [32]. Despite the significant success, GNNs may suffer from fairness issues due to biases inherited from training data and further amplified by their message-passing mechanism [1, 8, 9]. In the context of fairness in GNNs, a well-studied and popular research problem is group fairness which highlights algorithmic decisions that do not discriminate against or favor certain groups defined by the sensitive attribute, such as race and gender. In other words, group fairness aims to ensure that the outputs of GNNs are independent of the sensitive attribute.

In recent years, literature has been carried out to improve the group fairness of GNNs by mitigating biases from training data [9, 27, 38] or training GNNs with fairness-aware frameworks [5, 8, 47]. The core idea behind most of these approaches is removing the sensitive attribute-related information, thereby enforcing GNNs to make decisions independent of the sensitive attribute. However, such approaches inevitably remove some task-related information due to its correlation with the sensitive attribute, as shown in Figure 1(a). As a result, this leads to performance degradation of GNNs in downstream tasks. Although prior works also emphasize the trade-off between fairness and utility, it remains challenging to compensate for the performance degradation caused by removing task-related information.

To improve fairness while preserving task-related information, it is crucial to reasonably handle task-related information that is also associated with the sensitive attribute. Inspired by disentangled representation learning (DRL) [31, 33], disentanglement may be an established solution and provides valuable insights into fairness. DRL, whose goal is to recover a few explanatory factors of variation which generate the real-world data we observe, has

been proven to be beneficial for learning fair representation in Euclidean data [7, 30]. Despite its success, the effectiveness on graph-structured data remains under-explored. There may be two potential advantages for DRL in graph fairness: First, DRL reduces correlations between the sensitive attribute and other sensitive attribute-irrelevant representation dimensions due to its goal of learning independent representations for different latent factors [30, 35]. Secondly, DRL simplifies downstream prediction tasks and leads to better utility performance [42].

In light of DRL, we propose FairSAD, a simple yet effective graph representation learning framework for improving fairness while preserving task-related information. FairSAD has two key modules, i.e., (1) sensitive attribute disentanglement (SAD) and (2) sensitive attribute masking. SAD aims to learn disentangled node representations that the sensitive attribute is disentangled into independent components. Sensitive attribute masking involves weakening the sensitive attribute in task-related information via channel masking. As shown in Figure 1(b), the core idea behind FairSAD is the use of SAD, which mitigates the impact of the sensitive attribute-related information on other representation channels while preserving task-related information that is relevant to the sensitive attribute. In this way, FairSAD benefits from the independent representation characteristics of DRL with a higher fairness level. Simultaneously, it leverages the characteristic of capturing latent factors, leading to better utility performance. Our contributions are as follows:

- We explore a phenomenon wherein removing sensitive attribute-related information to improve fairness inadvertently leads to the removal of task-related information.
- We propose FairSAD, a graph representation learning framework for improving fairness while preserving utility. To our best knowledge, this is the first work to improve fairness in graph-structured data via disentanglement.
- We conduct extensive experiments on five real-world datasets, demonstrating the superior performance of FairSAD over other state-of-the-art fairness methods.

## 2 RELATED WORK

In this section, we give a brief overview of related work on disentangled representation learning and fairness in graph.

### 2.1 Disentangled Representation Learning

Disentangled representation learning, which learns representations that disentangle latent factors behind data, has emerged as a major area of research on representation learning [4, 31]. The prevalent technique of DRL on Euclidean data is derived from variational autoencoders (VAEs) [21], e.g., $\beta$-VAE [16], FactorVAE [20], $\beta$-TCVAE [6], WAEs [37]. Meanwhile, previous studies have shown that disentangled representations are highly promising for enhancing the generalization ability [44], robustness [2], and interpretability [28] of models.

Motivated by DRL on Euclidean data, research on graph-structured data also involves learning disentangled representations with GNNs. DisenGCN [33] is the first work to investigate the topic of disentangled representations learning on graph. DisenGCN proposes a neighborhood routing mechanism to identify the latent factor

causing the connection from a given node to one of its neighbors. To further improve DRL on graph, IPGDN [29], DGCF [45] build upon DisenGCN and further encourage independent representations between different latent factors through a regularizer. DisenKGAT [48] leverages a relation-aware aggregation mechanism and mutual information minimization to disentangle factors from micro- and macro-disentanglement views. Additionally, there are several alternative approaches available in disentangled graph representation learning, such as FactorGCN [49], DiCGRL [23], LGD-GCN [14]. The superior performance of DRL on graph has paved the way for the utilization of disentangled representation in diverse fields, including citation generation [46], molecule generation [10], recommendation [24]. Despite the great progress, the effectiveness of disentangled graph representation on fairness remains under-explored even if some studies on Euclidean data have shown that disentanglement may be a useful property to encourage fairness [30]. Thus, our work aims to explore learning fair graph representation via disentanglement.

### 2.2 Fairness in Graph

Fairness notions in graph can be divided into three categories, i.e., group fairness [11], individual fairness [19], and other fairness [34]. Group fairness, which emphasizes algorithmic decisions neither favor nor harm certain groups defined by the sensitive attribute, is investigated in this work. As an emerging area of research, prior studies have explored various approaches to improve group fairness. Adversarial learning-based approaches, such as FairGNN [8], FairVGNN [47], Graphair [27], aim to learn node representation or modify original data that fool discriminator to identify the sensitive attribute.

Another line of research balances node representation differences between multiple demographic groups divided by the sensitive attribute through some effective technologies, including minimizing distribution distance [9, 12, 54], reducing connection within the same demographic group [38], sampling neighbors with balance awareness [26], re-weighting edges to balance message [25]. Additionally, other approaches, e.g., Fairwalk [36], NIFTY [1], are empirically proven as effective approaches in improving group fairness. Despite these achievements, the core idea behind most fairness approaches is removing sensitive attribute-related information. Owing to such a design, these approaches inevitably remove some task-related information due to its correlation with the sensitive attribute, resulting in sacrificing utility. In contrast to previous works, FairSAD is designed to improve group fairness while preserving task-related information using disentanglement, an aspect that has remained under-explored in previous advances.

## 3 PRELIMINARY

In this section, we first introduce detailed notations in this paper, followed by the problem definition of this work.

### 3.1 Notations

In this paper, we focus on learning fair node representations. Let $\mathcal{G} = (\mathcal{V}, \mathcal{E}, \mathbf{X})$ denote an undirected attributed graph, comprised of a set of $|\mathcal{V}| = n$ nodes $\mathcal{V}$ and a set of $|\mathcal{E}| = m$ edges $\mathcal{E}$. $\mathbf{X} \in \mathbb{R}^{n \times d}$ represents the node attribute matrix where $d$ is the node attribute

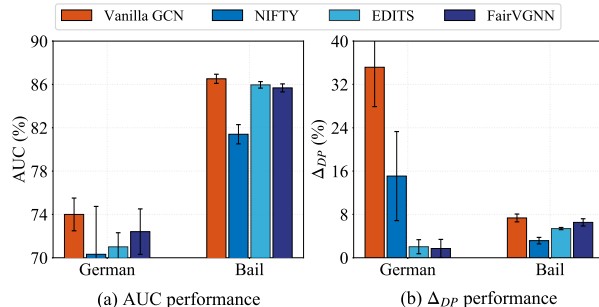

**Figure 2: Overview of FairSAD. FairSAD disentangles the sensitive attribute-related information into an independent component via sensitive attribute disentanglement, then reduces the correlation between the sensitive attribute and node representations via sensitive attribute masking. Disentangled layers in this example have three channels due to assuming three latent factors.**

dimension. $\mathbf{A} \in \{0, 1\}^{n \times n}$ is the adjacency matrix where $\mathbf{A}_{uv} = 1$ indicates that there exists edge $e_{uv} \in \mathcal{E}$ between the node $u$ and the node $v$, and $\mathbf{A}_{uv} = 0$ otherwise. $\mathbf{s} \in \{0, 1\}^n$ represents the binary sensitive attribute. For node $u$ and node $v$, $s_u = s_v$ indicates that these two nodes belong to the same demographic group. Most GNNs update the node representation vector $h$ through aggregating messages of its neighbors, which can be summarized as two steps: (1) message propagation and aggregation; (2) node representation updating. Thus, the $l$-th layer of GNNs is formalized as follows:

$$\mathbf{h}_u^{(l)} = \text{UPD}^{(l)}(\{\mathbf{h}_u^{(l-1)}, \text{AGG}^{(l)}(\{\mathbf{h}_v^{(l-1)} : v \in \mathcal{N}(u)\})\}), \quad (1)$$

where $\text{AGG}^{(l)}(\cdot)$ and $\text{UPD}^{(l)}(\cdot)$ denote aggregation function and update function in $l$-th layer, respectively. $\mathcal{N}(u)$ denote the set of nodes adjacent to node $u$.

Assume that each node representation consists of $K$ independent components corresponding to $K$ latent factors, i.e., $K$ channels. $\mathbf{h}_u = [\mathbf{z}_u^1, \mathbf{z}_u^2, ..., \mathbf{z}_u^k]$ is the disentangled representation vector of node $u$, where $\mathbf{z}_u^k \in \mathbb{R}^{\frac{d_h}{K}}$ ($1 \le k \le K$) is the $k$-th independent component, $d_h$ is the hidden dimension. Meanwhile, $\mathbf{H} = [\mathbf{Z}^1, \mathbf{Z}^2, ..., \mathbf{Z}^k]$ is the disentangled representations of a set of nodes, where $\mathbf{Z}^k$ is the $k$-th independent component of a set of nodes.

### 3.2 Problem Definition

Given $\mathcal{G}$ with the sensitive attribute $\mathbf{s}$, our goal is to learn fair node representation $\tilde{\mathbf{H}}$ while preserving task-related information. Take the node classification task as an example, the goal is to learn node representation for predicting node labels $\hat{\mathbf{y}}$. Here, $\hat{\mathbf{y}}$ should be independent of $\mathbf{s}$ while maintaining high classification accuracy.

## 4 PRESENT WORK: FAIRSAD

In this section, we discuss how to improve fairness while preserving task-related information with FairSAD. Specifically, we first give a detailed description of FairSAD, followed by optimization objectives. Figure 2 presents an overview of FairSAD. FairSAD consists of two modules, i.e., sensitive attribute disentanglement and sensitive attribute masking. In SAD, FairSAD disentangles the sensitive attribute into independent components to mitigate its impact

on other components. In sensitive attribute masking, FairSAD employs a channel masking to identify the sensitive attribute-related component while decorrelating it.

### 4.1 Sensitive Attribute Disentanglement

We investigate the performance of state-of-the-art graph fairness methods and vanilla GCN, as shown in Figure 3. We observe that despite significant progress in fairness performance ($\Delta_{DP}$), all methods inevitably experience a degradation in utility performance. A potential explanation for this phenomenon could be that the previous advances aim to eliminate the sensitive attribute information, inadvertently leading to the unintentional removal of the task-related information due to its correlations with the sensitive attribute.

To address the above issue, we propose improving fairness through SAD. This process involves separating sensitive attribute information into an independent component. When a latent factor corresponds to the sensitive attribute, the sensitive attribute-related information will be disentangled into the independent component, thereby alleviating its impact on other components. Meanwhile, the independent component facilitates further enhancements in fairness in the subsequent steps. Enjoying such a design, SAD also preserves task-related information related to the sensitive attribute.

**Figure 3: Performance of state-of-the-art fairness methods. Despite significant progress in fairness, these methods inevitably suffer from utility performance degradation.**

Specifically, SAD consists of two parts, i.e., neighbor assigner and disentangled layers. The former identifies the latent factor causing the connection between two nodes, while the latter performs multi-channel graph convolution to obtain disentangled representations.

*4.1.1 Neighbor Assigner.* To separate the sensitive attribute into an independent component, we need to identify latent factors leading to connection between nodes. DisenGCN [33] identifies latent factors in an iterative way and further obtains disentangled node representations. However, such a way has a high time complexity and iterative aggregation hinders the mining of latent factors. Instead of the iterative way, we propose a neighbor assigner to identify the latent factor causing the connection and pave the way for disentanglement.

Let $w_{uv}^k$ denote the edge weight from node $v$ to node $u$ in channel $k$, indicating the probability that latent factor $k$ causes the connection from node $v$ to node $u$. Here, channel $k$ corresponds to $k$-th latent factor. Our neighbor assigner is a multi-layer perceptron (MLP) and takes attributes of two connected nodes as input to predict the edge weight in different channels. Specifically, given the original attributes $\mathbf{x}_u \in \mathbb{R}^d$ of node $u$ and its neighbor $v$'s attributes $\mathbf{x}_v \in \mathbb{R}^d$, our neighbor assigner $f_a(\cdot)$ takes the concatenation of $\mathbf{x}_u$ and $\mathbf{x}_v$ as input to measure the importance of edge $e_{uv}$ for each latent factor. Then, we obtain $w_{uv}^k$ through the softmax operation:

$$\boldsymbol{\alpha}_{uv} = f_a([\mathbf{x}_u, \mathbf{x}_v]), w_{uv}^k = \frac{\exp(\alpha_{uv}^k)}{\sum_{j=1}^K \exp(\alpha_{uv}^j)}, \quad (2)$$

where $\boldsymbol{\alpha}_{uv} = [\alpha_{uv}^1, \alpha_{uv}^2, ..., \alpha_{uv}^k], (1 \leq k \leq K)$ is the importance score vector of $e_{uv}$ corresponding to $K$ latent factors.

Based on edge weights in $K$ channels, disentangled layers are capable of conducting multi-channel graph convolution to disentangle the sensitive attribute. Additionally, all edges in graph share parameters of the neighbor assigner, indicating the global and local level view of our neighbor assigner.

*4.1.2 Disentangled layers.* Combining edge weights in $K$ channels, we employ disentangled layers to perform graph convolution in multi-channel. A disentangled layer consists of $K$ channels graph convolution with the same network structure and each channel corresponds to a distinct latent factor. For a disentangled layer, the processing can be summarized as two steps: (1) *Dimensions Reduction*: reducing the dimension of the original attribute and projecting attributes into different subspaces corresponding to latent factors. (2) *Multi-channel Graph Convolution*: following the message-passing mechanism, aggregating and updating node representation based on edge weights predicted by the neighbor assigner.

For dimensions reduction, we use a linear layer as our reduction operation RED$(\cdot)$. Given $\mathbf{x}_u$ and the $k$-th reduction operation RED$_k(\cdot)$, we project $\mathbf{x}_u$ into the reduced node attribute $\mathbf{r}_u^k = $ RED$_k(\mathbf{x}_u), \mathbf{r}_u^k \in \mathbb{R}^{\frac{d_h}{K}}$. Here, we obtain $K$ reduced node attribute through $K$ independent reduction operation RED$(\cdot)$. For multi-channel graph convolution, we follow the widely used message-passing mechanism in GNNs, as shown in Eq.(1). Taking $\mathbf{z}_u^{k,(0)} = \mathbf{r}_u^k$, $\mathbf{z}_v^{k,(0)} = \mathbf{r}_v^k$, and $w_{uv}^k$ as input, the $l$-th layer of graph convolution in the $k$-th channel is formalized as follows:

$$\mathbf{z}_u^{k,(l)} = \text{UPD}_k^{(l)}(\{\mathbf{z}_u^{k,(l-1)}, \text{AGG}_k^{(l)}(\{w_{uv}^k \mathbf{z}_v^{k,(l-1)} : v \in \mathcal{N}(u)\})\}), \quad (3)$$

where $\text{UPD}_k^{(l)}(\cdot)$ and $\text{AGG}_k^{(l)}(\cdot)$ are the $k$-th update and aggregation function in the $l$-th layer, respectively. Note that our disentangled layer appears similar to GAT [43], but its purpose and actual functionality are entirely different. We utilize a *sum* operator as our aggregation function, i.e., computing the elementwise summarization of the vectors in $\{w_{uv}^k \mathbf{z}_v^{k,(l-1)} : v \in \mathcal{N}(u)\}$. To ensure numerical stability, we use $l_2$-norm to handle the aggregated and updated results, i.e., $\mathbf{z}_u^{k,(l)} = \mathbf{z}_u^{k,(l)}/\left\|\mathbf{z}_u^{k,(l)}\right\|_2$, thereby generating the final representation $\mathbf{z}_u^{k,(l)} \in \mathbb{R}^{\frac{d_h}{K}}$.

For notation simplicity, we omit the superscript "$(l)$" of the disentangled node representation below, i.e., $\mathbf{z}_u^{k,(l)}$ denote as $\mathbf{z}_u^k$. Concatenating all $\mathbf{z}_u^k$, we obtain the disentangled node representation of node $u$ in the $l$-th layer, denoted by $\mathbf{h}_u = [\mathbf{z}_u^1, \mathbf{z}_u^2, ..., \mathbf{z}_u^k]$. For the disentangled representation of all nodes, $\mathbf{Z}^k$ denote the disentangled representations of all nodes in $k$-th channel and $\mathbf{H} = [\mathbf{Z}^1, \mathbf{Z}^2, ..., \mathbf{Z}^k]$ is the concatenation of node disentangled representations for all channels. Assume that each latent factor channel consists of three columns, $\mathbf{H}$ can be transformed into a general representation with $d_h$ hidden dimensions, as follows:

$$\mathbf{H} = [\underbrace{\mathbf{c}_1, \mathbf{c}_2, \mathbf{c}_3}_{\mathbf{Z}^1}, ..., \underbrace{\mathbf{c}_{d_h-2}, \mathbf{c}_{d_h-1}, \mathbf{c}_{d_h}}_{\mathbf{Z}^k}] \quad (4)$$

where $\mathbf{c}_i \in \mathbb{R}^n, \forall i \in \{1, 2, ..., d_h\}$ is the disentangled representation of column feature form.

However, the above process only considers disentanglement at the sample level, neglecting the independence among latent factors, particularly the mutual independence across different channels. To tackle this issue, we incorporate distance correlation [40] and a channel discriminator [52] into the optimization objectives of Fair-SAD. These will be elaborated upon in the optimization objectives section.

## 4.2 Sensitive Attribute Masking

Assuming perfect sensitive attribute disentanglement, where sensitive attributes are completely disentangled into an independent component, we can focus solely on processing the independent component corresponding to the sensitive attribute to further improve fairness. In this regard, the problem is transformed into the sensitive attribute-related component identification and the sensitive attribute decorrelation for this component. Inspired by FairVGNN [47], which masks original features to generate fair views, we utilize a learnable channel mask $f_m$ to identify the independent component associated with the sensitive attribute while reducing the impact of the sensitive attribute. Unlike FairVGNN [47], our masking process takes the disentangled node representation $\mathbf{H}$ as input. Formally, the channel masking can be defined as follows:

$$\begin{aligned} \tilde{\mathbf{H}} = \mathbf{H} \odot \mathbf{m} &= [\mathbf{c}_1 m_1, \mathbf{c}_2 m_2, \mathbf{c}_3 m_3, ..., \mathbf{c}_{d_h} m_{d_h}] \\ &= [\underbrace{\tilde{\mathbf{c}}_1, \tilde{\mathbf{c}}_2, \tilde{\mathbf{c}}_3}_{\tilde{\mathbf{Z}}^1}, ..., \underbrace{\tilde{\mathbf{c}}_{d_h-2}, \tilde{\mathbf{c}}_{d_h-1}, \tilde{\mathbf{c}}_{d_h}}_{\tilde{\mathbf{Z}}^k}], \end{aligned} \quad (5)$$

where $\mathbf{m} = [m_1, m_2, ..., m_{d_h}], \mathbf{m} \in \mathbb{R}^{d_h}$ is the mask, $\tilde{\mathbf{H}}$ is the final node representation for downstream tasks.

A natural way to form the mask $\mathbf{m}$ is to sample following Bernoulli distribution, i.e., $m_i \sim \text{Bernoulli}(p_i)$, $\forall i \in \{1, 2, ..., d_h\}$. Here, $p_i$ denotes a learnable probability to mask a column feature channel. In this regard, we can learn a channel mask $f_m$. Due to the discreteness of masks, this process is non-differentiable. Thus, we approximate the Bernoulli distribution through the Gumbel-Softmax trick [17, 47]. The covariance constraint has been employed in [8] to minimize the absolute covariance between the noisy sensitive attribute and label prediction to achieve fairness. In our issue, we aim to learn a mask to adaptively identify the sensitive attribute-related component and decorrelate it. Thus, we regard the absolute covariance between the sensitive attribute $\mathbf{s}$ and each masked column feature $\tilde{\mathbf{c}}_i$, $\forall i \in \{1, 2, ..., d_h\}$ as loss function of $f_m$:

$$\mathcal{L}_m = \sum_{i=1}^{d_h} |Cov(\mathbf{s}, \tilde{\mathbf{c}}_i)| = \sum_{i=1}^{d_h} |\mathbb{E}[(\mathbf{s} - \mathbb{E}(\mathbf{s}))(\tilde{\mathbf{c}}_i - \mathbb{E}(\tilde{\mathbf{c}}_i))]|, \quad (6)$$

where $Cov(\cdot)$ is the covariance, $|\cdot|$ is the absolute value, $\mathbb{E}(\cdot)$ is the expectation operation. Overall, the core idea behind $f_m$ is to assign the minimal mask value to the sensitive attribute-related component.

## 4.3 Optimization Objectives

The goal of FairSAD is to learn fair graph representation $\tilde{\mathbf{H}}$ while preserving task-related information. In this regard, the optimization objectives of FairSAD can be divided into three parts: (1) *Downstream Tasks*: Objectives for downstream tasks ensure that the learned representation is informative and task-related. (2) *Disentanglement*: Objectives for disentanglement ensure the independence between latent factors. (3) *Decorrelation*: Objectives for decorrelation weaken the impact of the sensitive attribute-related component on final predictions. The loss function can be defined as follows:

*4.3.1 Downstream Tasks.* Take the node classification task as an example, the optimization objective $\mathcal{L}_c$ for the node classification task is a binary cross-entropy function.

*4.3.2 Disentanglement.* SAD already captures informative component representation corresponding to different factors, named micro-disentanglement [53]. To fully disentangle the sensitive attribute into an independent component, macro-disentanglement [53], emphasizing independence between different components, need to be considered in SAD. Thus, we employ distance correlation as a regularizer and a channel discriminator $f_d$ as a supervisor to guide the training of FairSAD.

Distance correlation [40], which is a measurement of dependence between random vectors, can characterize both the linear and non-linear relation. Thus, the loss function can be defined as follows:

$$\mathcal{L}_{dc} = \sum_{k_1=1}^{K} \sum_{k_2=k_1+1}^{K} \frac{dCov^2(\tilde{\mathbf{Z}}^{k_1}, \tilde{\mathbf{Z}}^{k_2})}{\sqrt{dCov^2(\tilde{\mathbf{Z}}^{k_1}, \tilde{\mathbf{Z}}^{k_1}) dCov^2(\tilde{\mathbf{Z}}^{k_2}, \tilde{\mathbf{Z}}^{k_2})}}, \quad (7)$$

where $dCov^2(\cdot)$ is the distance covariance and detailed calculation is shown in Appendix A.

The channel discriminator $f_d$, which is a linear layer in this work, takes $\tilde{\mathbf{Z}}^k$ as input and predicts the channel $\hat{y}'_{v,k}$ of each node. Naturally, we construct the label $y'_{v,k}$ using the channel index of $\tilde{\mathbf{Z}}^k$. Assume that $\mathcal{V}_T$ is a set of labeled nodes for training, we utilize

the cross-entropy function as the loss function, as follows:

$$\mathcal{L}_d = -\frac{1}{|\mathcal{V}_T|} \sum_{v \in \mathcal{V}_T} \sum_{k=1}^{K} y'_{v,k} \log \hat{y}'_{v,k}. \quad (8)$$

Note that we also utilize Eq.(8) to separately optimize $f_d$. Based on Eq.(7) and (8), FairSAD can learn the disentangled representation considering both micro- and macro-disentanglement, thereby separating the sensitive attribute-related information into an independent component.

*4.3.3 Decorrelation.* Decorrelation is achieved through the channel mask $f_m$, which aims to assign the minimum masking value to the sensitive attribute-related component. Our goal is to minimize correlations between the masked column features $\tilde{\mathbf{c}}_i$ and the sensitive attribute $\mathbf{s}$. Thus, the loss function is shown in Eq. (6).

Assume that we have the neighbor assigner $f_a$, the disentangled GNNs backbone $f_g$ with dimension reduction and $l$-layer disentangled layer, the channel mask $f_m$, and the classifier $f_c$, the optimization objectives of FairSAD can be summarized as follows:

$$\min_{\theta} \mathcal{L} = \mathcal{L}_c + \alpha(\mathcal{L}_{dc} + \mathcal{L}_d) + \beta\mathcal{L}_m, \quad (9)$$

where $\theta = \{\theta_{f_a}, \theta_{f_g}, \theta_{f_m}, \theta_{f_c}\}$ is the parameter set of $f_a$, $f_g$, $f_m$, and $f_c$. $\alpha$ and $\beta$ are hyperparameters that balance the contribution of disentanglement and decorrelation. To easily understand our proposed method FairSAD, we present a training algorithm of FairSAD and a forward propagation algorithm of the disentangled layer in Appendix B.

## 4.4 Theoretical Analysis

In this subsection, we present the theoretical analysis to prove why the proposed method FairSAD learns fair graph representations $\tilde{\mathbf{c}}_i$. Given an undirected attributed graph $\mathcal{G} = (\mathcal{V}, \mathcal{E}, \mathbf{X})$, we can obtain the node disentangled representations $\mathbf{H} = [\mathbf{Z}^1, \mathbf{Z}^2, ..., \mathbf{Z}^k]$ through sensitive attribute disentanglement in Section 4.1. $\mathbf{H}$ exhibits the sensitive attribute independence property, as shown in the following lemma.

LEMMA 1. *Assuming full disentanglement, where each channel representation $\mathbf{H}$ is independent of the others, it follows that at most one channel representation $\mathbf{Z}^k$ is related to the sensitive attribute $\mathbf{s}$.*

PROOF. Applying the proof by contradiction, we assume the existence of multiple channel representations related to the sensitive attribute. This implies correlations between channel representations related to sensitive attributes, which contradicts our initial premise of full disentanglement. Consequently, we conclude that at most one channel representation $\mathbf{Z}^k$ is related to $\mathbf{s}$. □

According to Lemma 1, the sensitive attribute-related information is separated into an independent component, which eliminates the impact of the sensitive attribute on other channel representations $\mathbf{Z}^k$. This constitutes the initial step in FairSAD towards improving fairness. Given the disentangled representation $\mathbf{H}$, we obtain the final node representations $\tilde{\mathbf{H}}$ through the sensitive attribute masking. We optimize the learnable mask using $\mathcal{L}_m$. In this context, we can prove that minimizing $\mathcal{L}_m$ helps our mask learn to identify the sensitive attribute-related component adaptively,

**Table 1: Statistic information of five real-world datasets.**

| Dataset | German | Bail | Credit | Pokec-z | Pokec-n |
|---------|--------|------|--------|---------|---------|
| #Nodes | 1,000 | 18,876 | 30,000 | 67,796 | 66,569 |
| #Edges | 22,242 | 321,308 | 1,436,858 | 617,958 | 583,616 |
| #Attr. | 27 | 18 | 13 | 277 | 266 |
| Sens. | Gender | Race | Age | Region | Region |

resulting in further improvements in fairness. The details of the proof are as follows:

PROPOSITION 1. *Let $\tilde{\mathbf{c}}_s$ denote the sensitive attribute-related channel representation in $\tilde{H}$, minimizing $\mathcal{L}_m$ is minimizing the correlation between the sensitive attribute $\mathbf{s}$ and $\tilde{\mathbf{c}}_s$.*

PROOF. We assume that $s$-th channel representation is the sensitive attribute-related component. Based on Lemma 1, we have the channel representations $\mathbf{Z}^k, k \neq s$ independent of the sensitive attribute $\mathbf{s}$. Thus, we also have final representations $\tilde{\mathbf{c}}_i, \forall i \in \{1, 2, ..., d_h\}, i \neq s$ in $\tilde{H}$ independent of the sensitive attribute. Then, Eq.(6) can be summarized as follows:

$$\mathcal{L}_m = \sum_{i=1}^{d_h} |Cov(\mathbf{s}, \tilde{\mathbf{c}}_i)| = |Cov(\mathbf{s}, \tilde{\mathbf{c}}_s)| = |\mathbb{E}[(\mathbf{s} - \mathbb{E}(\mathbf{s}))(\tilde{\mathbf{c}}_s - \mathbb{E}(\tilde{\mathbf{c}}_s))]|,$$

(10)

According to the above Equation, minimizing $\mathcal{L}_m$ is minimizing the correlation between the sensitive attribute $\mathbf{s}$ and $\tilde{\mathbf{c}}_s$. In this regard, the optimized mask can identify the sensitive attribute-related component adaptively. Meanwhile, a minimal mask value will be assigned to the sensitive attribute-related component due to minimizing the correlation between $\mathbf{s}$ and $\tilde{\mathbf{c}}_s$. □

## 5 EXPERIMENTS

In this section, we conduct experiments on five real-world datasets, whose statistics are shown in Table 1. We answer the following two research questions through experiments: (1) How effectively will FairSAD improve fairness? (2) What degree of utility performance can FairSAD maintain in the downstream task?

### 5.1 Experimental Settings

In this subsection, we give a brief overview of experimental settings. More details about this are shown in Appendix C.

*5.1.1 Datasets.* We conduct experiments on five commonly used datasets, including German, Bail, Credit [1], Pokec-z, and Pokec-n [8]. The statistics of datasets is shown in Table 1.

*5.1.2 Evaluation Metrics.* We use AUC and F1 scores to evaluate the utility performance. To evaluate fairness, we use two commonly used fairness metrics, i.e., $\Delta_{DP} = |P(\hat{y} = 1|s = 0) - P(\hat{y} = 1|s = 1)|$ [11] and $\Delta_{EO} = |P(\hat{y} = 1|y = 1, s = 0) - P(\hat{y} = 1|y = 1, s = 1)|$ [15]. $\hat{y}$ and $y$ denote the node label prediction and ground truth, respectively. For $\Delta_{DP}$ and $\Delta_{EO}$, a smaller value indicates a fairer model prediction.

*5.1.3 Baselines.* We compare the performance of FairSAD with five baseline methods, i.e., EDITS [9], Grahair [27], NIFTY [1], FairGNN [8], and FairVGNN [47].

*5.1.4 Implementation Details.* We conduct all experiments 5 times and reported average results. For a fair comparison, we tune hyperparameters for all methods according to the performance on the validation set. For FairSAD, we use a 1-layer disentangled layer with hidden dimensions $d_h = 16$ and set $K = 4$ for all datasets. We set $\alpha$, $\beta$ as $\{0.1, 0.001, 0.5, 0.001, 0.05\}$, $\{1.0, 0.2, 0.1, 0.05, 0.001\}$ for German, Bail, Credit, Pokec-z, and Pokec-n datasets, respectively. For all baselines, we follow the searching approach in [47] to select the best configuration of hyperparameters. More details about this are shown in Appendix C.3.

### 5.2 Comparison Study

We compare the performance of FairSAD with five baseline methods and vanilla GCN on the node classification task. Apart from FairSAD, all baseline methods utilize a 1-layer GCN as the backbone. This is because FairSAD is built upon our customized backbone, known as disentangled layers. Table 2 summarizes the comparison results of FairSAD with all baseline methods on the node classification task. We observe that FAirSAD outperforms all baseline methods across all evaluation metrics in most cases.

Notably, FairSAD demonstrates superior fairness performance (i.e., $\Delta_{DP}$ and $\Delta_{EO}$), as evidenced by the significant margin over all baseline methods across all datasets. This enhanced fairness can be attributed to two primary reasons: (1) SAD disentangles the sensitive attribute-related information into an independent component, alleviating its impact on other components. (2) Our channel masking mechanism reduces the influence of the sensitive attribute-related component on the final predictions. Simultaneously, Fair-SAD excels in utility performance, surpassing other methods in most cases. This outcome suggests the preservation of task-related information. Two potential explanations support these results: (1) FairSAD enhances fairness by eliminating the influence of the sensitive attribute on other components and weakening the impact of the sensitive attribute-related component on final predictions. Such a design facilitates the preservation of task-related information related to sensitive attributes. (2) Enjoying the advantage of disentanglement, FairSAD captures latent factors behind data, which simplifies downstream prediction tasks and results in better utility performance. To sum up, the experimental results demonstrate the effectiveness of FairSAD in improving fairness while preserving task-related information.

### 5.3 Ablation Study

We conduct ablation studies to gain insights into the effect of each component of FairSAD on improving fairness. Specifically, we denote FairSAD without SAD and sensitive attribute masking as "Fair-SAD w/o D" and "FairSAD w/o M", respectively. Note that FairSAD w/o D sets $K = 1$ and removes $f_a$, $\mathcal{L}_{dc}$, and $\mathcal{L}_d$. Table 3 presents ablation results on Credit, Pokec-z, and Pokec-n datasets. We observe that FairSAD performs better than two ablation variants on both fairness and utility performance, indicating the effectiveness of SAD and sensitive attribute masking. In Credit and Pokec-z datasets, FairSAD w/o M performs worse than FairSAD w/o D on both fairness and utility performance. Conversely, the opposite holds on the Pokec-n dataset. This result demonstrates that as the dataset undergoes changes, the contributions of SAD and sensitive

**Table 2: Comparison results of FairSAD with baseline fairness methods. In each row, the best result is indicated in bold, while the runner-up result is marked with an underline. OOM represents out-of-memory on a GPU with 24GB memory.**

| Datasets | Metrics | GCN | EDITS | Graphair | NIFTY | FairGNN | FairVGNN | FairSAD |
|---|---|---|---|---|---|---|---|---|
| German | AUC ($\uparrow$) | 65.90 ± 0.83 | 69.89 ± 3.23 | 47.03 ± 4.34 | 67.77 ± 4.30 | 67.35 ± 2.13 | **72.38 ± 1.09** | 70.39 ± 2.04 |
| | F1 ($\uparrow$) | 77.32 ± 1.20 | 82.01 ± 0.91 | 82.27 ± 0.23 | 81.43 ± 0.54 | 82.01 ± 0.26 | 81.94 ± 0.26 | **82.30 ± 0.10** |
| | $\Delta_{DP}$($\downarrow$) | 36.29 ± 4.64 | 2.38 ± 1.36 | 0.56 ± 1.11 | 2.64 ± 2.25 | 3.49 ± 2.15 | 1.44 ± 2.04 | **0.25 ± 0.51** |
| | $\Delta_{EO}$($\downarrow$) | 31.35 ± 4.39 | 3.03 ± 1.77 | 0.17 ± 0.29 | 2.52 ± 2.88 | 3.40 ± 2.15 | 1.51 ± 2.11 | **0.02 ± 0.04** |
| Bail | AUC ($\uparrow$) | 87.13 ± 0.31 | 87.92 ± 1.83 | 65.85 ± 20.61 | 79.62 ± 1.80 | 87.27 ± 0.76 | 87.05 ± 0.39 | **88.53 ± 0.62** |
| | F1 ($\uparrow$) | 78.98 ± 0.67 | 79.45 ± 1.48 | 55.00 ± 29.25 | 67.19 ± 2.63 | 77.67 ± 1.33 | **79.56 ± 0.29** | 78.22 ± 0.71 |
| | $\Delta_{DP}$($\downarrow$) | 9.18 ± 0.59 | 8.03 ± 0.97 | 4.33 ± 3.55 | 3.52 ± 0.72 | 6.72 ± 0.60 | 6.31 ± 0.77 | **2.97 ± 2.22** |
| | $\Delta_{EO}$($\downarrow$) | 4.43 ± 0.37 | 5.80 ± 0.73 | 2.85 ± 2.55 | 2.82 ± 0.82 | 4.49 ± 1.00 | 5.12 ± 1.40 | **2.30 ± 2.12** |
| Credit | AUC ($\uparrow$) | 70.71 ± 7.72 | 69.85 ± 1.43 | 53.21 ± 5.83 | **72.16 ± 0.12** | 71.95 ± 1.43 | 71.34 ± 0.79 | 72.09 ± 1.03 |
| | F1 ($\uparrow$) | 84.24 ± 1.32 | 84.67 ± 1.59 | 72.98 ± 9.07 | 81.74 ± 0.04 | 81.84 ± 1.19 | 78.75 ± 0.47 | **87.36 ± 0.19** |
| | $\Delta_{DP}$($\downarrow$) | 10.19 ± 0.98 | 5.57 ± 1.73 | 7.59 ± 7.42 | 11.71 ± 0.04 | 12.64 ± 2.11 | 3.46 ± 2.19 | **2.50 ± 2.01** |
| | $\Delta_{EO}$($\downarrow$) | 9.48 ± 0.61 | 2.41 ± 2.36 | 7.83 ± 6.97 | 9.42 ± 0.04 | 10.41 ± 2.03 | 1.91 ± 0.92 | **1.19 ± 1.55** |
| Pokec-z | AUC ($\uparrow$) | **76.42 ± 0.13** | OOM | 65.63 ± 0.38 | 71.59 ± 0.17 | 76.02 ± 0.15 | 75.52 ± 0.06 | 76.33 ± 0.44 |
| | F1 ($\uparrow$) | 70.32 ± 0.20 | OOM | 63.71 ± 1.19 | 67.13 ± 1.66 | 68.84 ± 3.46 | **70.45 ± 0.57** | 69.03 ± 0.91 |
| | $\Delta_{DP}$($\downarrow$) | 3.91 ± 0.35 | OOM | 1.59 ± 0.85 | 3.06 ± 1.85 | 2.93 ± 2.83 | 3.30 ± 0.87 | **0.97 ± 0.59** |
| | $\Delta_{EO}$($\downarrow$) | 4.59 ± 0.34 | OOM | 1.80 ± 0.60 | 3.86 ± 1.65 | 2.04 ± 2.27 | 3.19 ± 1.00 | **1.40 ± 0.65** |
| Pokec-n | AUC ($\uparrow$) | **73.87 ± 0.08** | OOM | 64.20 ± 0.92 | 69.43 ± 0.31 | 73.49 ± 0.28 | 72.72 ± 0.93 | 73.74 ± 0.54 |
| | F1 ($\uparrow$) | **65.55 ± 0.13** | OOM | 55.63 ± 1.42 | 61.55 ± 1.05 | 64.80 ± 0.89 | 62.35 ± 1.14 | 63.33 ± 2.49 |
| | $\Delta_{DP}$($\downarrow$) | 2.83 ± 0.46 | OOM | 4.77 ± 2.85 | 5.96 ± 1.80 | 2.26 ± 1.19 | 4.38 ± 1.73 | **1.88 ± 1.52** |
| | $\Delta_{EO}$($\downarrow$) | 3.66 ± 0.43 | OOM | 4.20 ± 3.62 | 7.75 ± 1.53 | 3.21 ± 2.28 | 6.74 ± 1.87 | **2.95 ± 1.83** |

**Table 3: Ablation results on Credit, Pokec-z, and Pokec-n.**

| Datasets | Metrics | FairSAD w/o D | FairSAD w/o M | FairSAD |
|---|---|---|---|---|
| Credit | AUC ($\uparrow$) | 70.52 ± 1.15 | 68.81 ± 3.48 | **72.09 ± 1.03** |
| | F1 ($\uparrow$) | 85.83 ± 1.79 | 82.90 ± 3.70 | **87.36 ± 0.19** |
| | $\Delta_{DP}$($\downarrow$) | **2.44 ± 1.77** | 6.53 ± 4.65 | 2.50 ± 2.01 |
| | $\Delta_{EO}$($\downarrow$) | 2.10 ± 1.17 | 5.02 ± 4.13 | **1.19 ± 1.55** |
| Pokec-z | AUC ($\uparrow$) | 76.26 ± 0.39 | 75.76 ± 0.51 | **76.33 ± 0.44** |
| | F1 ($\uparrow$) | 67.82 ± 0.62 | **69.59 ± 0.82** | 69.03 ± 0.91 |
| | $\Delta_{DP}$($\downarrow$) | 1.90 ± 0.65 | 4.16 ± 1.81 | **0.97 ± 0.59** |
| | $\Delta_{EO}$($\downarrow$) | 1.38 ± 0.88 | 4.67 ± 1.93 | **1.40 ± 0.65** |
| Pokec-n | AUC ($\uparrow$) | 71.90 ± 1.61 | 73.02 ± 0.60 | **73.74 ± 0.54** |
| | F1 ($\uparrow$) | 61.58 ± 1.63 | **64.18 ± 1.92** | 63.33 ± 2.49 |
| | $\Delta_{DP}$($\downarrow$) | 5.79 ± 0.90 | 2.46 ± 1.25 | **1.88 ± 1.52** |
| | $\Delta_{EO}$($\downarrow$) | 7.92 ± 1.00 | 3.47 ± 2.25 | **2.95 ± 1.83** |

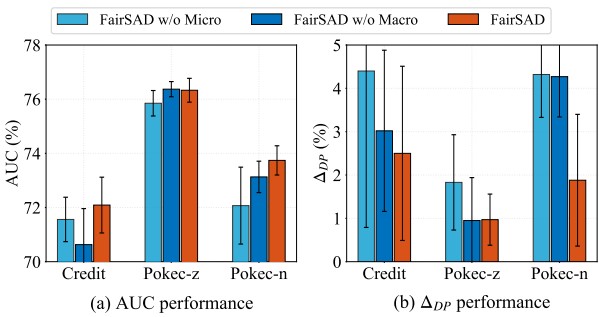

(a) AUC performance    (b) $\Delta_{DP}$ performance

**Figure 4: Ablation analysis w.r.t. micro- and macro-disentanglement on Credit, Pokec-z, and Pokec-n.**

attribute masking modules to the utility and fairness performance of FairSAD vary.

As shown in Figure 4, we further study the effect of micro- and macro-disentanglement on the performance of FairSAD. We remove $f_a$ and set $K = 1$, denoted as "FairSAD w/o Micro", and remove $\mathcal{L}_{dc}$, and $\mathcal{L}_d$, denoted as "FairSAD w/o Macro". We observe that FairSAD w/o Micro performs less effectively than FairSAD w/o Macro in terms of both utility and fairness performance, indicating that micro-disentanglement plays a more crucial role in enhancing the performance of FairSAD. Furthermore, this observation underscores the foundational importance of micro-disentanglement in the overall disentanglement process.

## 5.4 Parameters Sensitivity

We investigate the sensitivity of FairSAD w.r.t. two hyperparameters, i.e., $\alpha$ and $\beta$. In FairSAD, $\alpha$ and $\beta$ balance the contribution of disentanglement and decorrelation. Specifically, we vary the values of $\alpha$ and $\beta$ as $\{0.001, 0.01, 0.1, 0.5, 1, 5, 10\}$ on Credit, Pokec-z datasets. Figures 5 presents the results of the parameter sensitivity analysis. We make the following observation: (1) The overall performance of FairSAD remains stable across a wide range of variations in $\alpha$ and $\beta$. (2) When the values of $\alpha$ and $\beta$ are relatively small, the performance (F1 and $\Delta_{DP}$) of FairSAD is inferior compared to scenarios where the values of $\alpha$ and $\beta$ are relatively large. This observation highlights the equal contribution of disentanglement

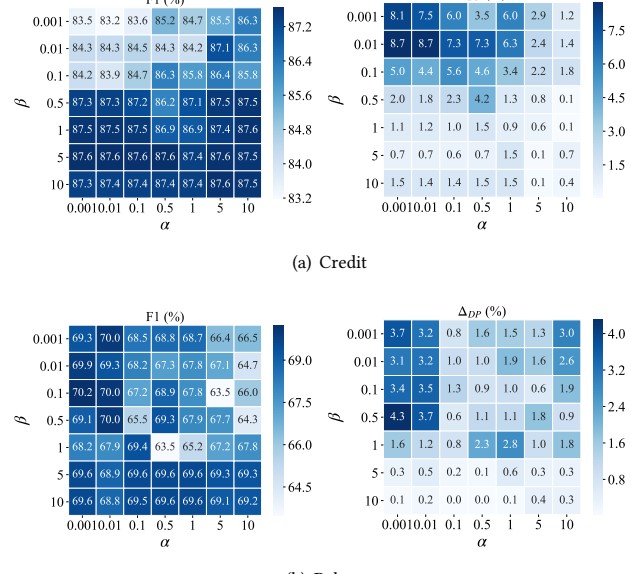

(a) Credit

(b) Pokec-z

Figure 5: Parameters sensitivity analysis on Credit and Pokec-z.

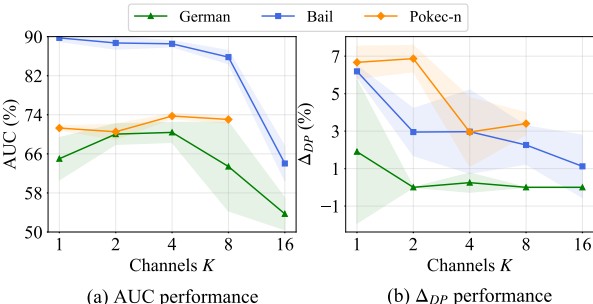

(a) AUC performance

(b) $\Delta_{DP}$ performance

Figure 6: Hyperparameter analysis w.r.t. channels $K$ on German, Bail, and Pokec-n.

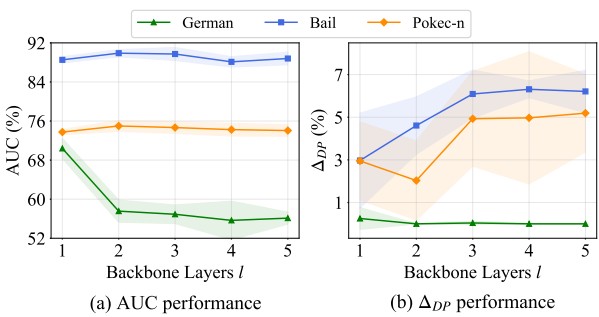

(a) AUC performance

(b) $\Delta_{DP}$ performance

Figure 7: Hyperparameter analysis w.r.t. backbone layers $l$ on German, Bail, and Pokec-n.

and decorrelation to FairSAD. To guarantee both the utility and fairness, $\alpha$ and $\beta$ are better to be selected from 0.01 to 1.0.

## 5.5 Further Probe

To further understand FairSAD, we conduct additional experiments to study the effect of channels $K$ and backbone layers $l$ on the performance of FairSAD. Due to out-of-memory on a GPU with 24GB memory, we only conduct additional experiments on German, Bail, and Pokec-n.

*5.5.1 Effect of Channels $K$.* We vary the value of $K$ as $\{1, 2, 4, 8, 16\}$ and fix other hyperparameters as the same as the comparison study. Figure 6 presents hyperparameter analysis results w.r.t. $K$. However, please note that Figure 6 does not include results for $K = 16$ on Pokec-n due to out-of-memory issues on a GPU with 24GB memory. We can observe that the utility (AUC) performance of FairSAD first increases and then decreases with the increase of channels $K$. This pattern suggests that a suitable $K$ (e.g., $K = 4$) may be the true number of latent factors underlying the data, resulting in the best utility performance. Meanwhile, as the increase of $K$, the fairness ($\Delta_{DP}$) performance improves. This demonstrates that a larger $K$ (e.g., $K \geq 4$) value is beneficial for disentangling the sensitive attribute, thereby mitigating its impact on other components. Overall, to balance the utility and fairness performance, selecting $K$ as 4 is preferable for our experimental datasets.

*5.5.2 Effect of Backbone Layers $l$.* We fix other hyperparameters as the same as the comparison study and vary the value of $l$ as $\{1, 2, 3, 4, 5\}$. Figure 7 presents hyperparameter analysis results w.r.t. $l$. We make the following observations: (1) FairSAD also suffers from performance degradation with the increase of the backbone

layer, indicating the over-smoothing issue of FairSAD. (2) As the number of layers increases, the fairness of FairSAD deteriorates. One potential explanation for this phenomenon is that the poor disentanglement performance caused by over-smoothing issues leads to a degradation in fairness.

## 6 CONCLUSION

In this paper, we study the problem of learning fair graph representations while preserving task-related information as much as possible. Inspired by disentangled representation learning, we explore addressing this problem through disentanglement. We propose a fair graph representation learning framework built upon disentanglement, namely FairSAD. The key insight behind FairSAD is to separate the sensitive attribute into the independent component via sensitive attribute disentanglement and then eliminate correlation with the sensitive attribute. Due to this design, FairSAD exhibits the advantage of improving fairness as well as preserving task-related information, resulting in superior performance on the downstream task. To our best knowledge, this is the first work to improve fairness in graph-structured data via disentanglement. Experiments on five real-world datasets demonstrate that FairSAD outperforms all baseline methods in terms of both fairness and utility performance.

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

## A  DISTANCE CORRELATION

Distance correlation [39, 40] measures the dependence between two random vectors. Inspired by DGCF [45], FairSAD employs distance correlation to ensure the independence between each component. According to Eq.(7), the distance correlation is built upon the distance covariance $dCov^2(\cdot)$. Given two component representations $\tilde{\mathbf{Z}}^{k_1}, \tilde{\mathbf{Z}}^{k_2} \in \mathbb{R}^{n \times \frac{d_h}{K}}$, we denote subscript $i, 1 \leq i \leq n$, and $j, 1 \leq j \leq \frac{d_h}{K}$ as the row, and column number, respectively. For instance, $\tilde{\mathbf{Z}}^{k_1}_{ij}$ represents the $i$-th row, $j$-th column of $\tilde{\mathbf{Z}}^{k_1}$. Formally, the distance covariance between two component representations can be defined as follows:

$$
\begin{aligned}
dCov^2(\tilde{\mathbf{Z}}^{k_1}, \tilde{\mathbf{Z}}^{k_2}) &= \sum_{j=1}^{\frac{d_h}{K}} dCov^2(\tilde{\mathbf{Z}}^{k_1}_{\cdot j}, \tilde{\mathbf{Z}}^{k_2}_{\cdot j}) \\
&= \sum_{j=1}^{\frac{d_h}{K}} \sum_{i_1=1}^{n} \sum_{i_2=1}^{n} A_{i_1 i_2} B_{i_1 i_2},
\end{aligned}
\tag{11}
$$

where $i_1, i_2 = 1, 2, ..., n$. To calculate $A_{i_1 i_2}$, we have the following equation:

$$
\begin{cases}
a_{i_1 i_2} = \|\tilde{\mathbf{Z}}^{k_1}_{\cdot j, i_1} - \tilde{\mathbf{Z}}^{k_1}_{\cdot j, i_2}\|_2 \\
\bar{a}_{i_1 \cdot} = \frac{1}{n} \sum_{i_2=1}^{n} a_{i_1 i_2} \\
\bar{a}_{\cdot i_2} = \frac{1}{n} \sum_{i_1=1}^{n} a_{i_1 i_2} \\
\bar{a}_{\cdot \cdot} = \frac{1}{n^2} \sum_{i_1, i_2=1}^{n} a_{i_1 i_2} \\
A_{i_1 i_2} = a_{i_1 i_2} - \bar{a}_{i_1 \cdot} - \bar{a}_{\cdot i_2} + \bar{a}_{\cdot \cdot}
\end{cases}
\tag{12}
$$

where $A_{i_1 i_2} \in \mathbb{R}^{n \times n}$. According to Eq.(12), we can calculate $B_{i_1 i_2}$ through defining $B_{i_1 i_2} = b_{i_1 i_2} - \bar{b}_{i_1 \cdot} - \bar{b}_{\cdot i_2} + \bar{b}_{\cdot \cdot}$. Following Eqs.(11) and (12), we can calculate $dCov^2(\tilde{\mathbf{Z}}^{k_1}, \tilde{\mathbf{Z}}^{k_1})$ and $dCov^2(\tilde{\mathbf{Z}}^{k_2}, \tilde{\mathbf{Z}}^{k_2})$. Based on the above three distance covariance, the distance correlation between $\tilde{\mathbf{Z}}^{k_1}, \tilde{\mathbf{Z}}^{k_2}$ can be calculated as shown in Eq.(7).

## B  ALGORITHM

To help better understand FairSAD, we present the training algorithm of FairSAD and the forward propagation algorithm of the disentangled layer in Algorithms 1 and 2. Note that the channel discriminator $f_d(\cdot)$ is solely employed during the training phase.

## C  EXPERIMENTAL SETTINGS

### C.1  Datasets

Five real-world fairness datasets, namely German, Bail, Credit [1], Pokec-z, and Pokec-n [8], are employed in our experiments. We give a brief overview of these datasets as follows:

---

**Algorithm 1** Taining Algorithm of FairSAD

**Input**: $\mathcal{G} = (\mathcal{V}, \mathcal{E}, \mathbf{X})$ with the sensitive attribute $\mathbf{s}$, node labels $\hat{\mathbf{y}}$, neighbor assigner $f_a(\cdot)$, disentangled GNNs backbone $f_g(\cdot)$, channel mask $f_m(\cdot)$, classifier $f_c(\cdot)$, channel discriminator $f_d(\cdot)$, channels number $K$, and hyperparameters $\alpha, \beta$.
**Output**: Trained inference GNNs model with parameters $\theta = \{\theta_{f_a}, \theta_{f_g}, \theta_{f_m}, \theta_{f_c}\}$.

1: **while** *not converged* **do**
2:     // Sensitive attribute disentanglement
3:     $\mathbf{W} \leftarrow f_a([\mathbf{x}_i, \mathbf{x}_j]), [\mathbf{x}_i, \mathbf{x}_j] = \{\mathbf{x}_i, \mathbf{x}_j \in \mathbf{X}, i, j \in \mathcal{V}, j \in \mathcal{N}(i)\}$;
4:     $\mathbf{H} \leftarrow f_g(\mathcal{G}, \mathbf{W})$;
5:     // Sensitive attribute masking
6:     $\tilde{\mathbf{H}} \leftarrow f_m(\mathbf{H})$;
7:     // Downstream tasks
8:     $\hat{\mathbf{y}} \leftarrow f_c(\tilde{\mathbf{H}})$;
9:     // Identifying channels using channel discriminator
10:     $\hat{\mathbf{y}}'_k \leftarrow f_d(\tilde{\mathbf{Z}}^k), \tilde{\mathbf{Z}}^k \in \tilde{\mathbf{H}}, \forall k \in \{1, 2, ..., K\}$;
11:     // Calculating loss and updating model parameters
12:     Calculate $\mathcal{L}_c, \mathcal{L}_m, \mathcal{L}_{dc}, \mathcal{L}_d$ following Eqs.(6), (7), and (8);
13:     Calculate loss function $\mathcal{L} \leftarrow \mathcal{L}_c + \alpha(\mathcal{L}_{dc} + \mathcal{L}_d) + \beta\mathcal{L}_m$;
14:     Update $\theta, \theta_{f_d}$ by gradient descent;
15: **end while**
16: **return** $\theta$;

---

**Algorithm 2** Forward Propagation Algorithm of the Disentangled Layer

**Input**: $\mathcal{G} = (\mathcal{V}, \mathcal{E}, \mathbf{X})$ with the sensitive attribute $\mathbf{s}$, edge weights $\mathbf{W} = [\mathbf{w}^1, \mathbf{w}^2, ..., \mathbf{w}^K]$, disentangled GNNs backbone $f_g(\cdot) = \{\text{RED}_k(\cdot), \text{UPD}_k(\cdot), \text{AGG}_k(\cdot)\}$.
**Output**: Disentangled node representations $\mathbf{H} = [\mathbf{Z}^1, \mathbf{Z}^2, ..., \mathbf{Z}^k]$.

1: // Sensitive attribute disentanglement
2: **for** $k = 1, 2, ..., K$ **do**
3:     $\mathbf{R}^k \leftarrow \text{RED}_k(\mathbf{X})$;
4:     $\mathbf{Z}^k \leftarrow$ Aggregate and update node representation through $\text{AGG}_k(\cdot)$ and $\text{UPD}_k(\cdot)$ following Eq.(3);
5: **end for**
6: // Sensitive attribute masking
7: $\mathbf{H} \leftarrow$ The concatenation of $\mathbf{Z}^1, \mathbf{Z}^2, ..., \mathbf{Z}^k$.
8: **return** $\mathbf{H}$;

---

- **German** [3] is constructed by [1]. Specifically, German includes clients' data in a German bank, e.g., gender, and loan amount. Nodes represent clients in the German bank. The edges in the German dataset are constructed according to individual similarity. Regarding "gender" as the sensitive attribute, the goal of German is to classify clients into two credit risks (high or low).
- **Bail** [1] is a defendants dataset, where defendants in this dataset are released on bail during 1990-2009 in U.S states [18]. We regard nodes as defendants and edges are decided by the similarity of past criminal records and demographics. Considering "race" as the sensitive attribute, the task is

to predict whether defendants will commit a crime after release (bail vs. no bail).

- **Credit** [1] is a credit card user dataset [50]. where nodes represent credit card users and edges are connected based on the similarity of their spending and payment patterns. Considering "age" as the sensitive attribute, the task is to predict whether a user will default on their credit card payment or not (default vs. no default).
- **Pokec-z/n** [8, 41] is derived from a popular social network application in Slovakia, where Pokec-z and Pokec-n are social network data in two different provinces. Nodes denote users with features such as gender, age, interest, etc. Edge represents the friendship between users. Considering "region" as the sensitive attribute, the task is to predict the working field of the users.

## C.2 Baselines

We compare FairSAD with five state-of-the-art fairness methods, including EDITS, Graphair, NIFTY, FairGNN, and FairVGNN. Among these five methods, EDITS and Graphair can be summarized as the pre-processing fairness method, which mitigates fairness-related biases in training data by modifying graph-structured data. NIFTY, FairGNN, and FairVGNN are the in-processing method, aiming to learn fair GNNs via the fairness-aware framework. A brief overview of these methods is shown as follows:

- **EDITS** [9] modify graph-structured data by minimizing the Wasserstein distance between two demographics.
- **Graphair** [27] aims to automatically generate fair graph data to fool the discriminator via adversarial learning.
- **NIFTY** [1] is a fair and stable graph representation learning method. The core idea behind NIFTY is learning GNNs to keep stable w.r.t. the sensitive attribute counterfactual.
- **FairGNN** [8] aims to learn fair GNNs with limited sensitive attribute information. To achieve this goal, FairGNN employs the sensitive attribute estimator to predict the sensitive attribute while improving fairness via adversarial learning.
- **FairVGNN** [47] learns a fair GNN by mitigating the sensitive attribute leakage using adversarial learning and weight clamping technologies.

## C.3 Implementation Details

For FairSAD, we utilize the Adam optimizer with the learning rate $lr = \{1 \times 10^{-3}, 1 \times 10^{-2}\}$, epochs = 1000, and the weight decay = $1 \times 10^{-5}$. For each component of FairSAD, we use a 2-layer MLP, a linear layer, and a linear layer as the neighbor assigner $f_a$, the classifier $f_c$, and the channel discriminator $f_d$, respectively.

For all baseline methods, we utilize a 1-layer GCN [22] with 16 hidden dimensions as the backbone and use a linear layer as the classifier. Due to the different hyperparameters for different baselines, we give the detailed setting for each baseline, as follows:

- **EDITS**: we set training epochs and threshold proportions as $\{500, 100, 500\}$, $\{0.2, 0.5, 0.02\}$ for German, Bail, and Credit. Initial learning rate $3 \times 10^{-3}$, weight decay $1 \times 10^{-7}$.

- **Graphair**: $\beta = 1$, $\alpha$, $\gamma$, and $\lambda$ are determined with a grid search among $\{0.1, 1, 10\}$, training epochs 500, initial learning rate $1 \times 10^{-4}$, weight decay $1 \times 10^{-5}$, adopt graph sampling-based batch training [51] with 1000 batch size for Bail, Credit, and Pokec-z/n.
- **NIFTY**: $\lambda = \{0.4, 0.6, 0.4, 0.6, 0.6\}$ for German, Bail, Credit, Pokec-z, and Pokec-n, training epochs 1000, learning rate $1 \times 10^{-3}$, weight decay $1 \times 10^{-5}$, drop edge rate 0.001, drop feature rate 0.1.
- **FairGNN**: dropout and learning rate are determined with a grid search among $\{0.0, 0.5, 0.8\}$, $\{0.0001, 0.001, 0.01\}$, weight decay $1 \times 10^{-5}$, $\alpha = 4$, $\beta = 0.01$.
- **FairVGNN**: training epochs for German, Bail, Credit, Pokec-z, and Pokec-n are $\{400, 300, 200, 200, 200\}$, other hyperparameters are determined with a grid search following search space set by the author.

Moreover, all experiments are conducted on one NVIDIA TITAN RTX GPU with 24GB memory. All models are implemented with PyTorch and PyTorch-Geometric.

Received 20 February 2007; revised 12 March 2009; accepted 5 June 2009

