# OpenReview forum: "Fair Graph Representation Learning via Sensitive Attribute Disentanglement"
_ACM.org/TheWebConf/2024/Conference — TheWebConf24 Oral_

### Official Review · Reviewer_W5tq · 2023-11-14

**Novelty:** 5
**Technical Quality:** 7

**Review:**

This paper proposes a two-pronged approach to learning fair GNNs, involving 1) sensitive attribute disentanglement and 2) sensitive attribute masking.

Pros

- Paper is in general written and organized clearly
- Interesting use of disentangled RL in graphs to promote fairness
- Marked improvements over baselines and extensive experiments that investigate the role of various hyperparameters
- Theoretical analysis is, to the best of my knowledge, sound

Cons

- Paper doesn’t connect the method’s significance to web applications (although it’s implicit that graph representations are highly applicable)

**Questions:**

Questions

- When $\alpha$ and $\beta$ are large, Figure 5 says that disparity (DP) is lowest which makes sense, but interestingly there is almost no dip in performance (F1) — in fact, performance in Credit seems to be monotonically increasing. I find that quite counterintuitive since higher $\alpha$ and $\beta$ means putting much less weight on $\mathcal{L}_c$. Do the authors have intuition as to why this is?
- It would be helpful to give a sense as to the maximal performance one could get without disentanglement and without considering fairness at all. It doesn’t have to be extremely involved, but showing this would help contextualize 1) how much performance is lost when enforcing fairness (if this is minimal, then it’s actually to the authors’ advantage), and 2) whether or not FairSAD’s gains relative to other methods is the “best” that a fairness-aware model could reasonably do. Right now, these improvements seem incremental (although improvements across the board nonetheless)

**Reviewer Confidence:**

3: The reviewer is confident but not certain that the evaluation is correct

**Scope:**

3: The work is somewhat relevant to the Web and to the track, and is of narrow interest to a sub-community

---

### Official Review · Reviewer_HrVu · 2023-11-23

**Novelty:** 5
**Technical Quality:** 5

**Review:**

In this paper, the authors present FairSAD, a novel framework aimed at improving fairness in Graph Neural Networks (GNNs) while preserving comparable prediction performance. FairSAD addresses the common issue of algorithmic bias in GNNs, where decisions may be influenced by sensitive attributes like race and gender. It innovatively employs Sensitive Attribute Disentanglement (SAD) to separate sensitive attribute-related information into an independent component, reducing its impact on decision-making. This approach contrasts with traditional methods that often compromise utility by eliminating sensitive information. Utilizing disentangled representation learning, FairSAD reduces correlations between sensitive and other data attributes, enhancing fairness without sacrificing the utility of the model. Extensive testing on five real-world datasets demonstrates that FairSAD outperforms existing state-of-the-art methods in both fairness and utility.

Strangth:

1.	FairSAD introduces an innovative approach by using Sensitive Attribute Disentanglement to separate sensitive attribute-related information. This method not only enhances fairness but also maintains the utility of GNNs

2.	The authors have rigorously evaluated FairSAD on five real-world datasets. This extensive testing not only demonstrates the framework's effectiveness but also underscores its applicability in practical scenarios.

3.	The paper effectively employs visualizations to present experimental results. This approach aids in conveying complex information clearly and succinctly, enhancing the comprehensibility of the findings.

Weakness:

1.	The paper could benefit from a more detailed explanation of how FairSAD restricts non-sensitive attributes that are associated with sensitive attributes. Merely isolating sensitive attributes might not completely nullify their influence. Clarification on identifying and handling non-sensitive attributes related to sensitive ones is necessary.

2.	The flowchart used in the paper lacks clear explanations, particularly regarding the meaning of the blue blocks, green squares, and red squares. A more detailed description of these elements would enhance the reader's understanding of the framework's process.

3.	FairSAD seems to overlook the aspect of graph reconstruction in its methodology. This oversight might lead to node embeddings that do not accurately represent the nodes, which could be a significant limitation in effectively capturing the graph structure's nuances.

**Questions:**

1.	It would be beneficial if the authors could further explain how FairSAD deals with non-sensitive attributes that are closely associated with sensitive attributes. Specifically, how does the framework identify and manage the impact of these non-sensitive attributes that might indirectly carry sensitive information?
2.	The paper would gain from a more detailed description of the flowchart elements, particularly the significance of the blue blocks, green squares, and red squares. Could the authors clarify the specific roles these elements play in the FairSAD framework? Additionally, it appears that graph reconstruction is not a central focus in the current methodology. How does this impact the representation accuracy of node embeddings and the overall graph structure?

**Reviewer Confidence:**

3: The reviewer is confident but not certain that the evaluation is correct

**Scope:**

4: The work is relevant to the Web and to the track, and is of broad interest to the community

---

### Official Review · Reviewer_vfLy · 2023-11-28

**Novelty:** 4
**Technical Quality:** 5

**Review:**

The authors propose to enhance fairness in graph-structured data through disentanglement representation learning and assert that their work is the first to explore this concept. Specifically, their proposed method, FairSAD, separates sensitive attribute-related information into an independent component through a process called sensitive attribute disentanglement. Subsequently, it diminishes the correlation between the sensitive attribute and node representations through sensitive attribute masking. The authors compared FairSAD with six other methods using five real-world datasets and achieved notable results. The paper is well-structured, easy to read, and presents an intriguing method. However, as I am not familiar with this topic, I am unable to provide an unbiased assessment of the paper's novelty.

**Questions:**

As the authors assert that the model is able to remove sensitive attribute-related information while retaining task-relevant information, my inquiry is focused on the identification process of distinguishing between sensitive information and task-pertinent information.

**Reviewer Confidence:**

2: The reviewer is willing to defend the evaluation, but it is likely that the reviewer did not understand parts of the paper

**Scope:**

4: The work is relevant to the Web and to the track, and is of broad interest to the community

---

### Official Review · Reviewer_kjrG · 2023-11-30

**Novelty:** 6
**Technical Quality:** 6

**Review:**

This paper proposes a group fairness approach for graph neural networks based on disentangled representations. The paper looks correct to me. It is a novel approach to the graph group fairness problem as far as I can tell. The experiments support the efficacy of the work.

The writing has several instances of redundancy where the same thing is said too many times in various parts of the document.

Typo on line 636: Grahair.

**Questions:**

-

**Ethics Review Description:**

-

**Reviewer Confidence:**

3: The reviewer is confident but not certain that the evaluation is correct

**Scope:**

4: The work is relevant to the Web and to the track, and is of broad interest to the community

---

### Decision · Program_Chairs · 2024-01-22

**Decision:**

Accept (Oral)

**Comment:**

Our decision is to accept. Please see the AC's review below and improve the work considering that and the reviewers' feedback for cemera-ready submission.

"Reviewers all found this a clear presentation of a novel idea, with nice results and evaluation. Some reviewers considered the work only tangentially related to the focus of TheWebConf, but I think this method has broad enough application that it will be of interest to many conference participants."